# Evaluating the Diagnostic Performance of Hemoglobin in the Diagnosis of Iron Deficiency Anemia in High-Altitude Populations: A Scoping Review

**DOI:** 10.3390/ijerph20126117

**Published:** 2023-06-13

**Authors:** Cinthya Vásquez-Velásquez, Daniel Fernandez-Guzman, Carlos Quispe-Vicuña, Brenda Caira-Chuquineyra, Fabricio Ccami-Bernal, Piero Castillo-Gutierrez, Miriam Arredondo-Nontol, Gustavo F. Gonzales

**Affiliations:** 1Laboratorio de Endocrinología y Reproducción, Laboratorios de Investigación y Desarrollo (LID), Departamento de Ciencias Biológicas y Fisiológicas, Facultad de Ciencias y Filosofía Alberto Cazorla Tálleri, Universidad Peruana Cayetano Heredia, Lima 15102, Peru; 2Instituto de Investigaciones de la Altura, Universidad Peruana Cayetano Heredia, Lima 15102, Peru; 3Carrera de Medicina Humana, Universidad Científica del Sur, Lima 15067, Peru; 4Sociedad Científica San Fernando, Universidad Nacional Mayor de San Marcos, Lima 15081, Peru; 5Facultad de Medicina, Universidad Nacional de San Agustín de Arequipa, Arequipa 04001, Peru; 6Escuela Profesional de Medicina Humana, Universidad Nacional de Tumbes, Tumbes 24001, Peru

**Keywords:** altitude, hemoglobin, iron deficiency anemia, scoping review, accuracy, ROC curves

## Abstract

We evaluated the available literature on the diagnostic performance of hemoglobin (Hb) in the diagnosis of iron deficiency anemia (IDA) in high-altitude populations. We searched PubMed, Web of Science, Scopus, Embase, Medline by Ovid, the Cochrane Library, and LILCAS until 3 May 2022. We included studies that evaluated the diagnostic performance (sensitivity, specificity, positive predictive value (PPV), negative predictive value (NPV), receiver operating characteristic (ROC) curves, and accuracy) of Hb (with and without an altitude correction factor) compared to any iron deficiency (ID) marker (e.g., ferritin, soluble transferrin receptor (sTFR), transferrin saturation, or total body iron (TBI)) in populations residing at altitudes (≥1000 m above sea level). We identified a total of 14 studies (with 4522 participants). We found disagreement in diagnostic performance test values between the studies, both in those comparing hemoglobin with and in those comparing hemoglobin without a correction factor for altitude. Sensitivity ranged from 7% to 100%, whereas specificity ranged from 30% to 100%. Three studies reported higher accuracy of uncorrected versus altitude-corrected hemoglobin. Similarly, two studies found that not correcting hemoglobin for altitude improved the receiver operating characteristic (ROC) curves for the diagnosis of iron deficiency anemia. Available studies on high-altitude populations suggest that the diagnostic accuracy of Hb is higher when altitude correction is not used. In addition, the high prevalence of anemia in altitude regions could be due to diagnostic misclassification.

## 1. Introduction

Anemia is considered a common problem of public health worldwide [1]. It affects mainly infants and children aged 6–59 months, women of reproductive age, pregnant women, and the elderly population, especially in developing countries [1]. Despite intensive interventions by different governments [2], the official statistics reveal that anemia prevalence has stagnated, and in low- and middle-income countries (LMICs) and developing countries, a trend to increase the prevalence of anemia has been observed [3].

The World Health Organization (WHO) recognizes iron deficiency (ID), mainly due to insufficient iron in the diet, followed by inflammation, as the main causes of anemia [4,5]. For such reason, almost all the governments in the world use iron supplementation and/or food fortification with iron as a strategy to reduce anemia prevalence. However, results are modest [2,6].

There are several explanations for this lack of response, including low adherence and the inefficacy of stakeholders to provide iron to the population, among others [7]. However, in recent years, other explanations have been given, such as the cutoff point to define anemia, but these are inadequate in different populations [8]. For example, the hemoglobin (Hb) cutoff for children aged 6–59 months is 11 g/dL, but this value does not reflect the normal change with age [9,10]. Another case is in pregnancy, in which the Hb cutoff is 1 g/dL lower in pregnant women, but this may not be enough for normal hemodilution in all pregnant women [11]. Finally, WHO recommends a correction of Hb per altitude as a universal rule for all the countries around the world [4]. However, not all populations living at the same altitude have similar Hb values, as observed in the Himalayas, where Tibetans have a lower Hb concentration than people of the Han ethnicity [12]. This is due to the fact that Tibetans have lived for more than 25,000 years in the highlands, whereas Hans have been living there for no more than 70 years [13,14].

Due to these reasons, it is necessary to properly establish the diagnosis of iron deficiency anemia (IDA) to avoid difficulties in the management and treatment of these patients. Therefore, in support of evidence-based decision making, we conducted this study with the aim of evaluating the available literature on the diagnostic accuracy of Hb in the diagnosis of IDA in high-altitude populations, since these populations have presented the highest prevalence of IDA when using Hb correction for altitude [15].

## 2. Materials and Methods

We performed a scoping review following the guidelines of the Preferred Reporting Items for Systematic and Meta-Analysis extension for Scoping Reviews (PRISMA-ScR) of 2018 [16] and the methodology described by the Joanna Briggs Institute [17]. A review protocol can be obtained from the Open Science Framework (https://osf.io/v7upm/ [accessed on 19 December 2022]).

### 2.1. Data Sources and Searches

We searched for articles in the following databases: (1) PubMed, (2) Web of Science/Core collection, (3) Scopus, (4) Embase, (5) Medline by Ovid, (6) Cochrane Library, and (7) Latin American and Caribbean Health Sciences Literature (LILCAS). The search was conducted on 3 May 2022. There were no restrictions regarding language or date of publication. Search terms were grouped into 4 categories: altitude, anemia for iron deficiency, Hb, and other hematological parameters of iron status. The full search can be found in Appendix A. 

### 2.2. Study Selection

We considered studies that evaluated the diagnostic performance of Hb (with and without a correction factor for altitude) in comparison with any ID marker (e.g., ferritin, soluble transferrin receptor (sTFR), transferrin saturation, or total body iron (TBI)) in populations residing at altitudes (≥1000 m above sea level). We also considered for study eligibility those studies that reported some diagnostic performance test, such as sensitivity, specificity, positive predictive value (PPV), negative predictive value (NPV), receiver operating characteristic (ROC) curves, and accuracy. If the data were sufficient to calculate a diagnostic performance test, they were also included.

Duplicate items were manually removed with Rayyan software [18]. Subsequently, two pairs of authors (FCB and CQV) independently reviewed the titles and abstracts of the studies to identify potentially relevant studies for inclusion. These studies were full-text-reviewed independently by two authors (MAN and PCG). Any disagreement in selection was discussed with another author (DFG) and resolved by consensus. In addition, the reference list of all included studies was reviewed to complement the search. Authors with expertise in the field were contacted for additional studies not included.

### 2.3. Data Extraction

An extraction sheet was designed in Microsoft Excel, and the data of interest were obtained independently by two authors (CQV and BCC). Discrepancies were resolved with a different author (DFG). First, we extracted report data, such as the title, publication date, first author, and publication type. Second, we extracted methodological data, such as the research design, setting, study period, countries, study population (infants, reproductive-age women, pregnant women, adults), and sample size. Third, we extracted information about the diagnosis of anemia and ID, such as the type of index test (Hb with or without the correction factor for altitude), the reference test (ferritin, soluble transferrin receptor (sTFR), transferrin saturation, total body iron (TBI), sTFR–ferritin index, plasma iron), and the cutoff points used for the diagnosis of anemia or ID. Finally, we extracted or calculated the prevalence of anemia by Hb, and ID by the reference test. The diagnostic performance of Hb (sensitivity, specificity, PPV, NPV, false positive rate, false negative rate, AUC, Youden index, and accuracy) was estimated. We did not plan to perform a formal critical appraisal of studies for this scoping review.

### 2.4. Data Synthesis 

We performed a synthesis of the diagnostic performance of Hb, grouping the studies by altitude of residence and age group (infants, reproductive-age women, pregnant women, adults). For this purpose, the studies were classified into those applying the same diagnostic procedures (e.g., studies comparing Hb to the same reference test), reference altitude (e.g., moderate altitude (1000 to 2500 m), high altitude (2500 to 3500 m), very high altitude (3500 to 5500 m), extreme altitude (>5500 m) [19]), and sample size. Finally, we presented the countries where the studies were performed by means of a map.

## 3. Results

### 3.1. Literature Search and Selection Process

In the systematic search, 312 studies were identified after removing duplicates. Of these, 79 full-text studies were reviewed for eligibility. Seven studies were additionally reviewed from other sources (expert recommendation and references of eligible studies). Finally, 14 studies were included in the review. The selection process is summarized in Figure 1. In addition, the reasons for exclusion of full-text studies reviewed are available in Appendix A.

### 3.2. Study Characteristics

The number of participants ranged from 66 to 804, with a total of 4522 high-altitude residents and with a predominance of women and children. Five studies evaluated only women of reproductive age [20,21,22,23,24], one study evaluated postpartum women [25], four studies included infants [24,26,27,28], and three studies evaluated the general population [29,30,31]. Regarding the altitude where the studies were conducted, we found six studies that included populations located between 1000 and 2500 m [20,21,22,32,33], four studies that included populations located between 2500 and 3500 m [25,26,29,30,32], and seven studies that evaluated populations located above 3500 m [23,24,27,28,29,30,32]. The characteristics of the studies are listed in Table 1.

Four studies were conducted in Bolivia [23,24,26,28] and three in Peru [27,32,32], while one study each was conducted in Saudi Arabia [20], South Africa [21], Ethiopia [22], Mexico [25], the Democratic Republic of the Congo [33], Papua New Guinea [31], and India [30] (Appendix A).

Of the 14 included studies, 3 [21,27,32] assessed the role of Hb in the diagnosis of anemia at an altitude using ROC curves. The reference standard (ID marker) most frequently evaluated was ferritin in 11 studies [20,21,22,24,25,26,27,28,30,31,33], followed by TBI in 4 studies [22,24,29,32], sTFR in 3 studies [21,22,24], and transferrin saturation in 3 studies [22,23,25]. The cutoff points for these markers were similar between studies according to the age of the population but with heterogeneity when adjusting for inflammation. With regard to the Hb cutoff point to classify as positive for anemia, we found 12 studies [20,21,22,24,25,26,27,28,29,31,32,33] used the correction for altitude recommended by WHO [4]; however, 6 studies [20,21,23,27,29,32] evaluated Hb without using this correction factor (Table 2 and Table 3). 

### 3.3. Diagnostic Performance Tests for IDA

The diagnostic performance of Hb versus other markers of iron status, including the prevalence of disease for each marker, are detailed in Table 2 and Table 3 and are briefly summarized in this section. The number of true and false positive cases and true and false negative cases for each comparator can be reviewed in Appendix A.

A. Hemoglobin vs. Ferritin, 11 studies.

The sensitivity of the studies ranged from 7% to 100%, the specificity ranged from 71% to 98%, and the accuracy ranged from 30% to 98%. In addition, Alkhaldy et al. reported that after correcting Hb for altitude (2270 m), the prevalence of anemia increased by 14%, well below the reported prevalence of ID [20]. Silubonde et al. (1700 m) found that a cutoff point higher than 12 g/dL of Hb (without correction for altitude) generated a higher discrimination threshold in the ROC curves for diagnosis of IDA [21]. In addition, Gonzales et al. [27] found that when not correcting Hb for altitude (3800 m), the prevalence of anemia and IDA by ferritin was similar, with a difference of 15% vs. ~70% when using the correction factor. The accuracy of Hb without the correction factor was between 63% and 69%. When the correction for altitude was used, the accuracy was between 31% and 77%. Only the study by Okumiya et al. [30] did not report whether the correction factor for altitude was used, and presented an accuracy of 83% in a population located between 2900 and 4900 m in India (Table 2).

B. Hemoglobin vs. sTFR, 3 studies.

The study by Burke et al. [24], performed at 4000 m, found a sensitivity of >90% for Hb, with the correction factor among infants (1 to 19 months) when the sTFR was corrected for inflammation status. In addition, it found a higher accuracy in older-age groups (6 to 19 months), as well as a lower specificity at older ages. However, Silubonde et al. [21] found that a value of 12.35 g/dL of Hb at an altitude of 1700 m for non-pregnant women obtained the highest threshold (ROC = 0.63) for the diagnosis of IDA. Finally, Gebreegziabher et al. [22] reported a sensitivity of 30%, a specificity of 79%, and an accuracy of 77% in non-pregnant women residing at 1708 m (Table 3).

C. Hemoglobin vs. TBI, 4 studies.

Alarcon-Yaquetto et al. [29] reported that the specificity and accuracy were higher for the diagnosis of IDA when the correction factor for altitude (3400 m) was not used. Gebreegziabher et al. [22] reported a higher diagnosis of anemia (21.3%) using Hb with the correction factor for altitude (1708 m) than when ID was evaluated with TBI (5.9%) in non-pregnant women of childbearing age. 

However, Choque-Quispe et al. [32] reported that in infants aged 6 to 59 months (altitude: 610–4660 m, only 4 children <1000 m), the accuracy decreased from 93% to 10% when the correction factor for altitude was used. They also found that the ROC had a greater area under the curve when Hb was not corrected, even after controlling for age, sex, and altitude. Finally, in the study by Burke et al. [24], which evaluated a cohort of newborns followed up to 10 to 19 months (4000 m), the authors found that the sensitivity of corrected Hb decreased with increasing age, while specificity and accuracy increased with increasing age (Table 3).

D. Hemoglobin vs. Transferrin saturation (TS), 3 studies.

Gebreegziabher et al. [22] reported in a group of non-pregnant women of childbearing age (1708 m) an accuracy of 73%. Villalpando et al. [25] reported a higher prevalence (62.1%) for Hb with correction versus the prevalence of ID measured by TS (47.0%) in postpartum women (2800 m). Meanwhile, Moreno-Black et al. [23] found that in non-pregnant women (3600 m), the accuracy was 93% when Hb was not corrected for altitude (Table 3).

E. Hemoglobin vs. other tests, 3 studies.

Finally, we highlight the study of Alarcon-Yaquetto et al. [29] that found in residents at 3400 m a higher accuracy for the diagnosis of IDA when the correction factor for altitude was not used (76% vs. 71%). Furthermore, the study by Gebreegziabher et al. [22] evaluated the performance of corrected Hb versus plasma iron (<500 μg/L), while Villalpando et al. compared Hb with a correction factor versus ≥2 indexes (serum ferritin (≤12 g/L), transferrin saturation (≤16%), or mean corpuscular value (MCV; ≤80 fL)); see Table 3.

## 4. Discussion

This review aimed to evaluate the diagnostic accuracy of Hb in IDA in high-altitude populations. We found 14 studies (with 4522 participants living at an altitude) that evaluated the diagnosis of IDA, comparing the value of Hb against a marker of ID. In general, we found heterogeneity in terms of age group, altitude of residence, and reference test to measure iron status. The prevalence of IDA was higher when only the Hb value was used but was lower when the correction for altitude was not used. Similarly, we found that the accuracy was higher for uncorrected Hb, mainly in populations above 3500 m. Regarding the assessment of the best diagnostic threshold (ROC curves), we found that in two studies [21,32], when Hb was not corrected for altitude, the AUC improved for making the diagnosis of IDA. As previous studies indicate, the high prevalence of anemia in high-altitude populations could be due to a systematic error in using a factor that does not have good diagnostic performance [11,34,35,36]. In addition, the different metabolism of high-altitude residents could cause a lower response rate to iron supplementation [37]).

In 1989, WHO recommended an adjustment for the residence altitude for Hb values for the diagnosis of anemia, starting the correction from an arbitrary point at 1000 m [4,38]. This cutoff point has a basis that is not representative worldwide and, to date, has several problems when used in some high-altitude populations. Thus, populations of greater generational seniority, such as the Ethiopians (Ethiopia) and Tibetans (Nepal, China, and India), have lower Hb than those of lesser generational seniority, due to a possible process of transgenerational genetic adaptation [12]. Similarly, in the Peruvian Andes, epigenetic modifications in DNA have been reported that could explain an adaptation of residents born in the region to altitude [39]. Furthermore, in the older population (Quechua residents of southern Peru), a lower Hb level has been observed than in the residents of central Peru, which would indicate a greater adaptation to altitude [40]. In this sense, the findings of the studies included in this review indicate that the diagnostic value of Hb in IDA in high-altitude residents is better when a correction factor is not considered. This would suggest a possible physiological adaptation process, mainly at altitudes above 3000 m, which would justify re-evaluating the anemia cutoff points for each region with high-altitude population, considering generational seniority.

We found six studies that used Hb not corrected for altitude in the evaluation of the diagnosis of IDA. We observed that not using correction for altitude decreased the prevalence of anemia by 5% to 83% [24,27,29,32], with the greatest drop in high-altitude populations. In this regard, a meta-analysis found that Hb levels increase as one ascends to higher altitudes and that the prevalence of anemia (with use of the correction factor) is the highest in the high Andean regions of the world [41]. In addition, based on population survey data, researchers have found that current recommendations for adjusting Hb levels may under-adjust Hb for occasional smokers and for those residing at lower altitudes and over-adjust Hb for those residing at higher altitudes [42].

Of greater concern is the overadjustment in Andean and Tibetan populations that present a different adaptation phenotype for altitude compared to other populations [43]. In the same sense, it has been reported that correction for altitude should not be used to assess the effects of maternal Hb on newborn outcomes, as this correction could minimize the true risk of presenting erythrocytosis and/or anemia at an altitude [44].

In clinical practice, Hb is used to diagnose anemia. Furthermore, it is common to observe ID as the first cause of anemia [1]. However, in high-altitude residents, problems have been reported in adequately identifying the causes of anemia. For example, in children in the Puno-Peru region (over 3800 m), of a total of 223 children diagnosed with anemia (using the correction factor), the authors estimated that about 22% were attributable to iron deficiency and 28% to inflammatory anemia, with 50% of children diagnosed with anemia but with no identifiable common cause [45]. This would suggest that there is an overestimation in the diagnosis of anemia. It should be recognized that other causes of anemia, such as heredity or infections, should be less prevalent than IDA and would not justify the high proportion of unidentified anemia [1]. Therefore, these findings would suggest that there is an inaccurate diagnosis of IDA in high-altitude residents, which could lead to overdiagnosis and inappropriate iron treatment. It is necessary to highlight that unnecessary treatment with ferrous sulfate could generate negative effects in some populations; for example, in pregnant women, high iron levels could reduce the utero-placental flow and more likely cause intrauterine growth retardation [46].

Our scoping review provides information about the diagnostic value of Hb with and without correction for altitude to predict the IDA status in high-altitude populations. We also discuss the current problem of using corrected cutoff points for the diagnosis of anemia at an altitude and the implications of misdiagnosing IDA. Based on this, we recommend that the established recommendations for diagnosing anemia in high-altitude residents be reevaluated. In addition, consideration should be given to the geographic and ethnic diversity and physiologic adaptations of residents with generational seniority. The use of different iron markers as the gold standard for diagnosing IDA should be used for work-up and research protocols to obtain a clearer and more comparable picture between studies, considering the adjustment of these tests for inflammatory conditions.

The main limitation of the included studies was the small sample size (with a minimum of 66 and a maximum of 492 participants included) from non-randomized enrollments. In addition, 13 of the 14 studies had an analytical cross-sectional design, limiting the understanding of variation over time in the baseline tests. However, the cutoff points and adjustment factors used for Hb and the reference tests were heterogeneous. Finally, only three studies aimed to assess the diagnostic threshold for Hb without a correction factor.

Similarly, our scoping review has some limitations. First, although the primary evidence was not entirely scarce, it was not uniform, as it was mostly concentrated in South American regions, which would prevent us from obtaining a global picture of our problem. Second, the methodological quality of the included studies was not formally evaluated, which would limit the reliability of the reported results. Finally, by including only published studies, there is a possibility of not having covered the totality of studies performed and incurring a publication bias. Despite these limitations, our study also has strengths. First, to the best of our knowledge, this is the first scoping review that provides an overview of the literature in this area. Second, this review was based on the methodological recommendations of the JBI [17] and PRISMA [16]. Finally, we believe that our results can contribute to the scientific evidence on the diagnostic value of Hb for the diagnosis of iron deficiency in high-altitude residents.

## 5. Conclusions

In summary, limited evidence is available on the diagnostic accuracy of Hb with and without a correction factor for altitude in the diagnosis of IDA. However, available studies in high-altitude populations suggest that the diagnostic accuracy of Hb is higher when altitude correction is not used. In addition, the high prevalence of anemia in high-altitude regions could be due to diagnostic misclassification. Based on this, we recommend that the established recommendations for diagnosing anemia in high-altitude residents be reevaluated, considering the geographic and ethnic diversity and physiological adaptations of residents with generational seniority. We also suggest further evaluation of the topic in order to facilitate a systematic review in the future.

## Figures and Tables

**Figure 1 ijerph-20-06117-f001:**
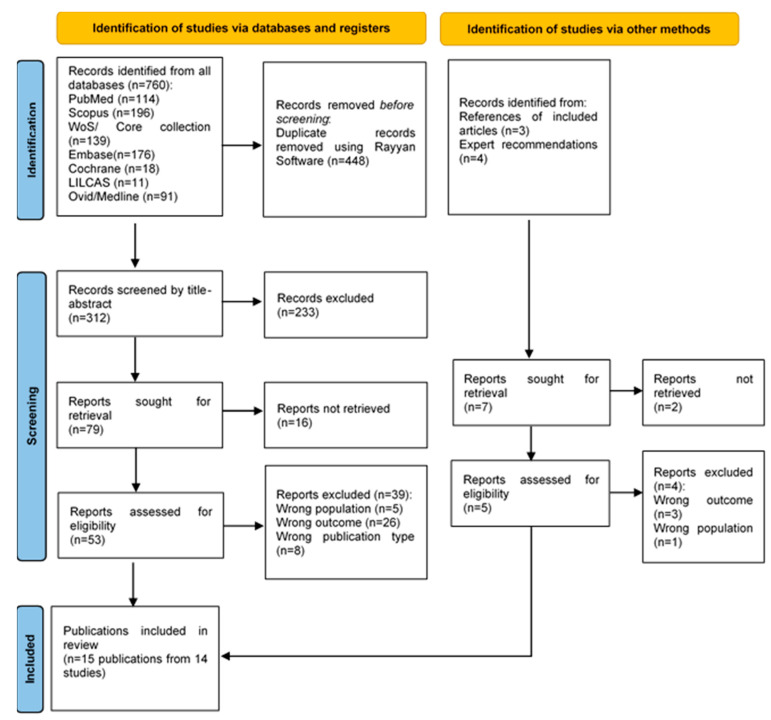
Flow diagram summarizing the process of literature search and selection.

**Table 1 ijerph-20-06117-t001:** Characteristics of diagnostic test accuracy of studies: 14 studies and 7 comparisons.

Source	Study Design	Country	Altitude (m)	Setting	Sponsor	Population				Reference Standard Evaluated (Marker of Iron Deficiency)
						Patient Population or Subpopulation	N	Age: Mean ± SD or Range	Males (%)	Type of Test
Alarcon-Yaquetto et al. (2022) [29]	Cross-sectional	Peru (Cusco)	3400	Community	Yes	Male or female over 18 years, born and living in Cusco	345	45.4 ± 15.6 years	45.2	sTFR–ferritin index (sTFR/log ferritin) and TBI
Alkhaldy et al. (2020) [20]	Cross-sectional	Saudi Arabia (Aseer—Abha)	2270	Community	No	Female medical students	200	20.4 ± 1.5 years	0.0	Ferritin
Choque-Quispe et al. (2020) [32]	Cross-sectional	Peru (Puno)	610–4660	Community	Yes	Children under 5 years (only 4 children living at <1000 m)	403	34.5 ± 1.5 months	48.9	TBI
Silubonde et al. (2020) [21]	Cross-sectional	South Africa (Gauteng)	1700	Community	No	Women of African descent, generally healthy and not pregnant	492	21 ± 3 years	0.0	Ferritin, sTFR
Gonzales et al. (2020) * [27]	Cross-sectional letters	Peru (Puno)	3800	Community	Yes	Infants	133	6–24 months	NR	Ferritin
Burke et al. (2018) [28]	Cross-sectional	Bolivia (El Alto)	4000	Hospital	Yes	Infants	270	7 ± 1 months	52.6	Ferritin
Bahizire et al. (2017) [33]	Cross-sectional	Democratic Republic of the Congo (Kivu—Miti-Murhesa)	1500–2000	Community	Yes	Infants from rural malaria-endemic areas	804	6–59 months	0.1	Ferritin
Burke et al. (2016) [24]	Cohort	Bolivia (El Alto)	4000	Hospital	Yes	Infants	160	1–5 months	53.1	Ferritin, sTFR, TBI
								6–10 months		
								10–19 months		
						Mothers of enrolled infants	250	25.3 ± 6.3 years	0.0	Ferritin, sTFR, TBI
Gebreegziabher et al. (2017) [22]	Cross-sectional	Ethiopia (Sidama)	1708	Community	Yes	Non-pregnant women over 18 years	202	30.8 ± 7.8 years	0.0	Ferritin, sTFR, transferrin saturation, TBI, plasma iron
Okumiya (2016) [30]	Cross-sectional * letter	India (Ladakh)	2900–4900	Community	Yes	Farmers and nomads from India and migrants from Tibet	500	55.9 ± 10.9 years	43.6	Ferritin
Miranda et al. (2015) [26]	Cross-sectional	Bolivia (Sucre)	2750	School	No	Schoolchildren from 6 to 10 years in the peri-urban area	195	8.3 ± 1.3 years	54.9	Ferritin
Villalpando et al. (2003) [25]	Cross-sectional	Mexico (Capulhuac)	2800	Hospital	No	Postpartum women	66	24 ± 5 years	0.0	Ferritin, transferrin saturation, serum iron
Moreno-Black et al. (1984) [23]	Cross-sectional	Bolivia (La Paz)	3600	Community	Yes	Women participating in the “Mothers’ Club” in Bolivia	152	33.8 ± 5.7 years	0.0	Transferrin saturation
Vaterlaws et al. (1981) [31]	Cross-sectional	Papua New Guinea (Goroka)	1500	Community	No	Rural areas	350	15–60 years	96.9	Ferritin

sTFR: soluble transferrin receptor; TBI: total body iron; NR: not reported. * The studies in question complement information from the same evaluated population.

**Table 2 ijerph-20-06117-t002:** Results of diagnostic test accuracy of studies—hemoglobin vs. ferritin.

Source	Patient Population	Hemoglobin with Factor Correction	Cutoff of Reference Standard	Prevalence of Anemia	Prevalence of Iron Deficiency	Se	Sp	PPV	NPV	FPR	FNR	ROC	Accuracy
				n (%)	n (%)								
A. Population: Adults over 18 years of age, not pregnant													
Alkhaldy et al. (2020) [20]	Female medical students from 19 to 27 years of age (n = 200); altitude 2270 m	Yes	<15 ng/mL	41 (20.5)	105 (52.5)	0.33	0.94	0.85	0.56	0.67	0.06	NR	0.62
		Yes	<20 ng/mL	41 (20.5)	126 (63.0)	0.3	0.96	0.93	0.45	0.7	0.04	NR	0.55
		No (12.00 g/dL)	<15 ng/mL	13 (6.5)	116 (58.0)	NR	NR	NR	NR	NR	NR	NR	NR
Silubonde et al. (2020) [21]	Women of African descent, generally healthy and not pregnant, between 18 and 25 years (n = 492); altitude 1700 m	No (12.45 g/dL)	<15 µg/L	192 (39)	178 (36.2)	0.58	0.71	0.52	0.76	0.42	0.29	0.68	0.67
		Yes	<15 µg/L *	192 (39)	185 (37.6)	0.57	0.72	0.55	0.73	0.43	0.28	NR	0.66
		No (12.00 g/dL)	<15 µg/L *	91 (18.5)	185 (37.6)	0.34	0.91	0.68	0.69	0.66	0.09	NR	0.69
		No (12.35 g/dL)	<15 µg/L *	183 (37.2)	185 (37.6)	0.56	0.74	0.56	0.73	0.44	0.26	0.68	0.67
		No (12.45 g/dL)	<30 µg/L	192 (39)	254 (51.6)	0.52	0.74	0.68	0.59	0.48	0.26	0.65	0.63
Gebreegziabher et al. (2017) [22]	Non-pregnant women over 18 years; altitude 1708 m	Yes	≤15 µg/L *	43 (21.3)	36 (17.8)	0.28	0.8	0.23	0.84	0.72	0.2	NR	0.71
Okumiya (2016) [30]	Farmers and nomads from India and migrants from Tibet; altitude 2900–4900 m	NR	≤12 ng/mL	59 (11.8)	116 (23.2)	0.4	0.97	0.78	0.84	0.6	0.03	NR	0.83
Vaterlaws et al. (1981) [31]	Adults in rural areas; altitude 1500 m	Yes	<30 ug/L	10 (2.9)	64 (18.3)	0.07	0.98	0.6	0.77	0.93	0.02	NR	0.77
B. Population: Postpartum women													
Villalpando et al. (2003) [25]	Postpartum women; altitude 2800 m	Yes	≤12 µg/L	41 (62.1)	35 (53)	NR	NR	NR	NR	NR	NR	NR	NR
C. Population: Children under 5 years of age													
Gonzales et al. (2020) [27]	Infants 6 to 24 months old; altitude 3800 m	Yes	≤12 ng/ml	126 (94.7)	35 (26.3)	0.54	1	NR	NR	0.46	0	0.68	0.58
		No (11.0 g/dL)	≤12 ng/ml	15 (11.3)	35 (26.3)	NR	NR	NR	NR	NR	NR	NR	NR
Burke et al. (2018) [28]	Infants; altitude 4000 m	Yes	≤12 µg/L *	204 (75.6)	151 (55.9)	0.83	0.34	0.61	0.61	0.17	0.66	NR	0.61
Bahizire et al. (2017) [33]	Infants 6 to 59 months old; altitude 1500–2000 m	Yes	≤12 ug/L	377 (46.9)	17 (2.1)	0.88	0.54	0.04	1	0.12	0.46	NR	0.55
		Yes	<30 ug/L *	377 (46.9)	82 (10.2)	0.76	0.56	0.16	0.95	0.24	0.44	NR	0.58
Burke R.M. et al. (2016) [24]	Infants 1–5 months old; altitude 4000 m	Yes	≤ 12 μg/L	112 (70.0)	1 (0.5)	1	0.3	0.01	1	0	0.7	NR	0.31
		Yes	≤ 12 μg/L *	112 (70.0)	1 (0.5)	1	0.3	0.01	1	0	0.7	NR	0.31
	Infants 6–10 months old; altitude 4000 m	Yes	≤ 12 μg/L	121 (75.6)	64 (40.3)	0.86	0.32	0.46	0.77	0.14	0.68	NR	0.54
		Yes	≤ 12 μg/L *	121 (75.6)	89 (55.5)	0.83	0.34	0.61	0.61	0.17	0.66	NR	0.61
	Infants 10–19 months old; altitude 4000 m	Yes	≤ 12 μg/L	130 (81.3)	105 (65.5)	0.88	0.31	0.71	0.57	0.12	0.69	NR	0.68
		Yes	≤ 12 μg/L *	130 (81.3)	127 (79.2)	0.85	0.35	0.83	0.39	0.15	0.65	NR	0.75
D. Population: Children from 6 to 10 years old													
Miranda M. et al. (2015) [26]	Schoolchildren from 6 to 10 years of age in the peri-urban area; altitude 2750 m	Yes	<30 μg/L *	35 (17.9)	38 (19.5)	0.44	0.99	0.97	0.73	0.56	0.01	NR	0.77

Se: sensitivity; Sp: specificity; PPV: positive predictive value; NPV: negative predictive value; FPR: false positive rate; FNR: false negative rate; ROC: receiver operating characteristic; NR: not reported; * adjusted for inflammation.

**Table 3 ijerph-20-06117-t003:** Results of diagnostic test accuracy of studies—hemoglobin vs. other markers of iron deficiency.

Source	Patient Population	Hemoglobin with Factor Correction	Cutoff of Reference Standard	Prevalence of Anemia	Prevalence of Iron Deficiency	Se	Sp	PPV	NPV	FPR	FNR	ROC	Accuracy
				n (%)	n (%)								
A. hemoglobin vs. sTFR													
Silubonde et al. (2020) [21]	Women of African descent, generally healthy and not pregnant (n = 492); altitude 1700 m	No (12.35 g/dL)	≥8.3 mg/L	183 (37.2)	204 (41.5)	NR	NR	NR	NR	NR	NR	0.633	NR
Gebreegziabher et al. (2017) [22]	Non-pregnant women over 18 years; altitude 1708 m	Yes	≥8.3 mg/L	43 (21.3)	10 (5)	0.3	0.79	0.07	0.96	0.7	0.21	NR	0.77
Burke et al. (2016) [24]	Infants 1–5 months old; altitude 4000 m	Yes	≥8.3 mg/L	112 (70.0)	2 (1.4)	0.57	0.3	0.01	0.98	0.43	0.7	NR	0.3
		Yes	≥8.3 mg/L *	112 (70.0)	1 (0.6)	1.28	0.3	0.01	1.01	−0.28	0.7	NR	0.31
	Infants 6–10 months old; altitude 4000 m	Yes	≥8.3 mg/L	121 (75.6)	13 (8.4)	0.96	0.26	0.11	0.99	0.04	0.74	NR	0.32
		Yes	≥8.3 mg/L *	121 (75.6)	12 (7.4)	0.96	0.26	0.09	0.99	0.04	0.74	NR	0.31
	Infants 10–19 months old; altitude 4000 m	Yes	≥8.3 mg/L	130 (81.3)	39 (24.4)	0.95	0.23	0.28	0.93	0.05	0.77	NR	0.4
		Yes	≥8.3 mg/L *	130 (81.3)	43 (26.8)	0.94	0.23	0.31	0.91	0.06	0.77	NR	0.42
B. hemoglobin vs. Transferrin saturation													
Gebreegziabher et al. (2017) [22]	Non-pregnant women over 18 years; altitude 1708 m	Yes	<15%	43 (21.3)	32 (15.8)	0.31	0.81	0.23	0.86	0.69	0.19	NR	0.73
Villalpando et al. (2003) [25]	Postpartum women; altitude 2800 m	Yes	≤16%	41 (62.1)	31 (47.0)	NR	NR	NR	NR	NR	NR	NR	NR
Moreno-Black et al. (1984) [23]	Women participating in the “Mothers’ Club” in Bolivia; altitude 3600 m.	No (13.0 g/dL)	<15%	6 (3.9)	17 (11.1)	0.35	1	1	0.92	0.65	0	NR	0.93
C. hemoglobin vs. TBI (mg/kg) = −[log10 (sTfR × 1000/ferritin) − 2.8229]/0.1207.													
Alarcon-Yaquetto et al. (2022) [29]	Male or female over 18 years, born and living in Cusco (n = 345); altitude 3400 m	No (12.0 for women and 13.0 g/dL for men)	<0 (mg/kg) *	2 (0.6)	0 (0.0)	0	0.99	0	0.77	1	0.01	NR	0.76
		Yes	<0 (mg/kg) *	21 (6.1)	0 (0.0)	0	0.86	0	0.74	1	0.14	NR	0.66
Choque-Quispe et al. (2020) [32]	Infants 6–59 months old (n = 403); altitude 610–4660 m	No (11.0 g/dL)	<0 (mg/kg) *	14 (3.5)	5 (2.5)	0.6	0.94	0.21	0.99	0.4	0.06	NR	0.93
		Yes	<0 (mg/kg) *	182 (45.2)	8 (4.1)	0.75	0.07	0.03	0.87	0.25	0.93	NR	0.1
	Infants 6–35 months old (n = 200); altitude 610–4660 m	No (11.0 g/dL)	<0 (mg/kg) *	12 (6.0)	NR	NR	NR	NR	NR	NR	NR	0.66 (0.41–0.90)	NR
		Yes	<0 (mg/kg) *	122 (61.2)	NR	NR	NR	NR	NR	NR	NR	0.62 (0.52–0.71)	NR
	Infants 36–59 months old (n = 203); altitude 610–4660 m	No (11.0 g/dL)	<0 (mg/kg) *	8 (3.9)	NR	NR	NR	NR	NR	NR	NR	0.88 (0.77–0.99)	NR
		Yes	<0 (mg/kg) *	138 (68.0)	NR	NR	NR	NR	NR	NR	NR	0.63 (0.54–0.72)	NR
	Infants 36–59 months old without inflammation (n = NR); altitude 610–4660 m	No (11.0 g/dL)	<0 (mg/kg) *	NR	NR	0.33	0.97	0.33	0.97	NR	NR	0.81 (0.79–0.84)	NR
		Yes	<0 (mg/kg) *	NR	NR	0.83	0.33	0.04	0.98	NR	NR	0.63 (0.62–0.64)	NR
Burke et al. (2016) [24]	Infants 1–5 months old; altitude 4000 m	Yes	<0 (mg/kg)	112 (70.0)	1 (0.6)	1	0.3	0.01	1	0	0.7	NR	0.31
		Yes	<0 (mg/kg) *	112 (70.0)	1 (0.6)	1	0.3	0.01	1	0	0.7	NR	0.31
	Infants 6–10 months old; altitude 4000 m	Yes	<0 (mg/kg)	121 (75.6)	44 (27.7)	0.93	0.31	0.34	0.92	0.07	0.69	NR	0.48
		Yes	<0 (mg/kg) *	121 (75.6)	62 (39.0)	0.91	0.34	0.47	0.86	0.09	0.66	NR	0.56
	Infants 10–19 months old; altitude 4000 m	Yes	<0 (mg/kg)	130 (81.3)	88 (54.8)	0.95	0.36	0.64	0.86	0.05	0.64	NR	0.68
		Yes	<0 (mg/kg) *	130 (81.3)	108 (67.3)	0.91	0.38	0.75	0.67	0.09	0.62	NR	0.74
Gebreegziabher et al. (2017) [22]	Non-pregnant women over 18 years; altitude 1708 m	Yes	<0 (mg/kg) *	43 (21.3)	12 (5.9)	NR	NR	NR	NR	NR	NR	NR	NR
D. Hemoglobin vs. sTFR–ferritin index = sTfR/log ferritin													
Alarcon-Yaquetto et al. (2022) [29]	Male or female over 18 years of age, born in the city of Cusco and living permanently in the city (n = 345); altitude 3400 m	No	TFR-F index < 1.0 and IL-6 < 50 pg/mL	2 (0.6)	12 (3.5)	0.02	0.99	0.5	0.76	0.98	0.01	NR	0.76
		Yes	TFR-F index < 1.0 and IL-6 < 50 pg/mL	21 (6.1)	12 (3.5)	0.19	0.93	0.52	0.73	0.81	0.07	NR	0.71
E. Hemoglobin vs. plasma iron													
Gebreegziabher et al. (2017) [22]	Non-pregnant women 18–52 years of age; altitude 1708 m	Yes	<500 μg/L	43 (21.3)	53 (26.2)	0.3	0.82	0.37	0.77	0.7	0.18	NR	0.68
F. Hemoglobin vs. ≥2 indexes (serum ferritin (≤12 g/L), transferrin saturation (≤16%), or MCV (≤80 fL))													
Villalpando et al. (2003) [25]	Postpartum women; altitude 2800 m	Yes	≥2 indexes	41 (62.1)	26 (39.4)	0.92	0.58	0.59	0.92	0.08	0.43	NR	0.71

sTFR: soluble transferrin receptor; TBI: total body iron; Se: sensitivity; Sp: specificity; PPV: positive predictive value; NPV: negative predictive value; FPR: false positive rate; FNR: false negative rate; ROC: receiver operating characteristic; NR: not reported; MCV: mean corpuscular volume; * adjusted for inflammation.

## Data Availability

All data are available in the paper and Appendix A.

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
