# Peer review of "Evaluating the Diagnostic Performance of Hemoglobin in the Diagnosis of Iron Deficiency Anemia in High-Altitude Populations: A Scoping Review"

_ijerph, 2023, doi:10.3390/ijerph20126117_

Round 1

Reviewer 1 Report

This is a well done study which challenges the current recommendation of adjusting Hb for altitude and this is very interesting. The methods were clear. I did notice quite a few grammatical errors (some of which i have pointed below) which need to be evaluated and corrected.

Line 42: "including in different countries in developed 42 and developing countries" . This needs grammatical correction 

Line 59: "living in a same altitude" should be "the" same altitude

Line 68: "Since this population has presented the highest prevalence when using the Hb correction for altitude (15).": prevalence "of anemia"

Line 281: Are there similar observations in adult patients? If yes, please summarize them here as well

Line 288, 289: "Therefore, these findings would suggest that there is an inaccurate diagnosis of IDA in high-altitude residents, which could lead to overdiagnosis and inappropriate iron treatment. " Can you explain how an overdiagnosis of anemia can lead to innapropriate treatment of IDA. Diagnosis of IDA requires looking at iron content in addition to Hb, so this statement is confusing. 

Author Response

Manuscript ID: ijerph-2182272

Title: Evaluating the diagnostic performance of hemoglobin in the diagnosis of iron deficiency anemia in high-altitude populations: A Scoping Review

Response to Reviewer 1 Comments

Comment Nº 1: This is a well done study which challenges the current recommendation of adjusting Hb for altitude and this is very interesting. The methods were clear. I did notice quite a few grammatical errors (some of which i have pointed below) which need to be evaluated and corrected.

Line 42: "including in different countries in developed 42 and developing countries" . This needs grammatical correction 

Response Nº 1: We appreciate your comment. The text has been modified; it has been placed in developed countries and Low and Middle Income Countries.

Comment Nº 2: Line 59: "living in a same altitude" should be "the" same altitude

Response Nº 2: We appreciate your comment. The text has been modified.

Comment Nº 3: Line 68: "Since this population has presented the highest prevalence when using the Hb correction for altitude (15).": prevalence "of anemia"

Response Nº 3: We appreciate your comment. The text has been modified.

Comment Nº 4: Line 281: Are there similar observations in adult patients? If yes, please summarize them here as well.

Response Nº 4: We appreciate your comment. In this geographical area of Peru there is evidence of children and infants.

Comment Nº 5: Line 288, 289: "Therefore, these findings would suggest that there is an inaccurate diagnosis of IDA in high-altitude residents, which could lead to overdiagnosis and inappropriate iron treatment. " Can you explain how an overdiagnosis of anemia can lead to innapropriate treatment of IDA. Diagnosis of IDA requires looking at iron content in addition to Hb, so this statement is confusing. 

Response Nº 5: We appreciate your comment. Further on, line 291-293, it is detailed that misdiagnosis leads to irrational iron supplementation, which can be harmful due to its toxicity.

Reviewer 2 Report

I was not familiar with scoping reviews, and have looked at some guidelines, I am still confused about their role. The question that is being addressed is not clear to me. Is it the prevalence of iron deficiency anaemia (IDA) in the heterogeneous populations who live at high altitude, or is it a comparison of the diagnostic performance of haemoglobin, with and without altitude-adjusted cutoffs? In my view there is little value in including papers that merely describe prevalence of anaemia or IDA (and related parameters) in a community (based on a single cutoff), since there are so many factors that affect the prevalence of IDA. The value to a reader is the comparison of different cutoffs in the context of the WHO recommendation for altitude adjustment. This is especially true for those studies that used non-standard methods for classification of ID (e.g., TBI, transferrin saturation, plasma iron).

Once the question is clarified, shouldn’t the scoping review then critically assess how many and which studies are able to address the question?  Looking at the description of the papers, it seems that very few (maybe just one) can answer the question of whether to adjust or not.  The authors have done some preliminary data analysis, but how much analysis belongs in a scoping review?  From a haematologist’s point of view I would have preferred to wait for a full analysis, rather than have this halfway point that is not critical enough to be meaningful.

I have annotated the text with a variety of comments (not comprehensive).

Author Response

Manuscript ID: ijerph-2182272

Title: Evaluating the diagnostic performance of hemoglobin in the diagnosis of iron deficiency anemia in high-altitude populations: A Scoping Review

Response to Reviewer 2 Comments

Comment Nº 1: I was not familiar with scoping reviews, and have looked at some guidelines, I am still confused about their role. The question that is being addressed is not clear to me. Is it the prevalence of iron deficiency anaemia (IDA) in the heterogeneous populations who live at high altitude, or is it a comparison of the diagnostic performance of haemoglobin, with and without altitude-adjusted cutoffs? In my view there is little value in including papers that merely describe prevalence of anaemia or IDA (and related parameters) in a community (based on a single cutoff), since there are so many factors that affect the prevalence of IDA. The value to a reader is the comparison of different cutoffs in the context of the WHO recommendation for altitude adjustment. This is especially true for those studies that used non-standard methods for classification of ID (e.g., TBI, transferrin saturation, plasma iron).

Response Nº 1: We appreciate your comment. Although systematic reviews and scoping reviews are related [1], scoping reviews differ from systematic reviews in several respects. Scoping reviews are used to present an overview of the evidence pertaining to a topic, regardless of the quality of the study, and are useful when examining emerging areas to clarify key concepts and identify gaps [2]. For example, scoping reviews can be used to identify a topic area for a future systematic review. Systematic reviews, on the other hand, are used to address more specific questions based on particular criteria of interest (i.e., population, intervention, outcome, etc.) defined a priori [2]. Scoping reviews can be considered a hypothesis-generating exercise, whereas systematic reviews can be hypothesis testing.

Since scoping reviews allow us to explore and better understand knowledge gaps on a topic, we decided to perform this methodology, since research to reevaluate a hemoglobin cutoff point in high-altitude population remains scarce, with heterogeneous populations and measurement procedures. In addition, since most of the literature is not focused on answering a PICO question, the present study aimed to synthesize the available evidence on the diagnostic evaluation of hemoglobin in the diagnosis of iron deficiency anemia, which could be a first approach to the need for research on this topic.

Comment Nº 2: Once the question is clarified, shouldn’t the scoping review then critically assess how many and which studies are able to address the question?  Looking at the description of the papers, it seems that very few (maybe just one) can answer the question of whether to adjust or not.  The authors have done some preliminary data analysis, but how much analysis belongs in a scoping review?  From a haematologist’s point of view I would have preferred to wait for a full analysis, rather than have this halfway point that is not critical enough to be meaningful.

 Response Nº 2: We appreciate your comment. Since a preliminary search prior to conducting the review did not detect any articles that could answer a PICO question to assess the diagnostic accuracy of hemoglobin in the high-altitude population, we decided to conduct a scoping review to more broadly address the information on the topic. In addition, due to the lack of studies focused on evaluating the diagnostic accuracy of hemoglobin with and without correction for altitude in the diagnosis of IDA, a systematic review would not be able to collect information from different types of study designs or with different comparators to define IDA in populations at different altitudes.

Comment Nº 3: I have annotated the text with a variety of comments (not comprehensive).

Response Nº 3: We appreciate your comments. All suggestions and corrections noted were accepted and implemented.

Reference

[1] Moher D, Stewart L, Shekelle P. All in the Family: systematic reviews, rapid reviews, scoping reviews, realist reviews, and more. Syst Rev. 2015;4(1):183.

[2] Peters MD, Godfrey CM, Khalil H, McInerney P, Parker D, Soares CB. Guidance for conducting systematic scoping reviews. Int J Evid Based Healthc. 2015;13(3):141–6.

Round 2

Reviewer 1 Report

No comments